# Body Mass Index: Influence on Interpersonal Style, Basic Psychological Needs, Motivation, and Physical Activity Intention in Physical Education—Differences between Gender and Educational Stage

**DOI:** 10.3390/bs13121015

**Published:** 2023-12-15

**Authors:** David Manzano-Sánchez, Alberto Gómez-Mármol, Manuel Gómez-López

**Affiliations:** 1Department of Didactics of Musical, Plastic and Corporal Expression, Faculty of Education and Psychology, University of Extremadura, 06006 Badajoz, Spain; davidms@unex.es; 2Department of Didactics of Plastic, Musical and Dynamic Expression, Faculty of Education, University of Murcia, 30100 Murcia, Spain; 3Campus of International Excellence “Mare Nostrum”, University of Murcia, 30100 Murcia, Spain; mgomezlop@um.es; 4Department of Physical Activity and Sport, Faculty of Sports Sciences, University of Murcia, 30720 Murcia, Spain

**Keywords:** physical activity, physical education, educational stage, gender differences, autonomy support

## Abstract

The present research study’s main objective was to find out whether there is a relationship between the body mass index (BMI) and the psychological aspects related to motivation, needs such as autonomy, competence, and social relationships, and the intention of being physically active in Physical Education students in Primary and Secondary Education. To achieve this, a total of 574 students (mean = 13.66; standard deviation = 1.96) participated in this study, to whom a series of questionnaires was administered once permission had been obtained from the centers the students attended, alongside the latter’s acceptance to participate in the study. The main results showed that the students with a higher BMI were those who had lower self-determined motivation values with regard to the three basic psychological needs observed and the intention of physical activity. In turn, our differential analysis identified that the students attending Primary Education had lower values of BMI, motivation, and intention to be physically active than the students attending Secondary Education, without finding differences based on the gender of the participants. The need to keep on investigating this topic is consequently gathered, using direct techniques for measuring BMI or proposing mixed research designs.

## 1. Introduction

The high rates of sedentary lifestyles and low levels of physical activity (PA) in children and adolescents translate into metabolic problems and chronic diseases [1]. To this are added the results of research that relate the body mass index (BMI) to cardiometabolic risk scores in both children [2] and adolescents [3]. Although there are several ways of calculating BMI values, the most widespread is based on dividing weight (measured in kilograms) by height, measured in meters, squared [4,5]. It is a recommended way to carry out measurements in large samples due to its ease and low cost [6], although it is known that it carries a certain risk of incorrectly classifying subjects, especially among the active population [7], because, for example, it does not distinguish between fat and non-fat tissue or the proportion of hydration and lean mass [8]. The BMI groups according to measured values have been established by the WHO as follows: a BMI < 18.5 is considered underweight; a BMI ≥ 18.5 and <25 is considered normal weight; a BMI ≥ 25 and <30 is considered overweight; and a BMI ≥ 30 is considered obese, although there are studies which suggest the adjustment of these cut-off points according to gender [9] and age [10].

With regard to a sedentary lifestyle, the World Health Organization (WHO) stated in 2020 [11] that 81% of children and adolescents did not comply with the PA recommendations proposed in 2010 [12]. Moreover, the WHO [13] recalls that this sedentary lifestyle is associated with greater adiposity, worse cardiometabolic health and physical fitness, and shorter sleep duration. The prevalence of sedentary lifestyle has increased [14,15], among other motives, because active leisure has, in many cases, been exchanged for passive leisure, thus reducing PA levels [16]. Therefore, a sedentary lifestyle, high BMI values, and a lack of PA are among the main health risks for young people; hence, the habits acquired during childhood are essential for long-term healthy growth [13]. The literature has shown that the subject of Physical Education (PE) can become a tool to combat this sedentary lifestyle, since it is an ideal context to promote active and healthy habits both inside and outside school [17,18]. As a consequence, the teacher is one of the most responsible agents involved in creating satisfactory experiences in the classroom, so his intervention becomes determinant to increasing motivation and satisfying the basic psychological needs (BPN) of students [19]. The problem justifying the importance of this research lies in the levels of sedentary lifestyles among young people and their consequences for health, not only for physiological health-related concerns (BMI) but also for psychological health issues as, for instance, basic psychological needs (BPN) or motivation modulate behaviors which lead to the integration of either healthy or unhealthy habits into daily routine practices.

According to Deci and Ryan [20], there are three BPNs that student must satisfy. The first of these is autonomy, which refers to the feeling that students have when they are the protagonists of what they do and are, thus, able to assess their own performance and make decisions when solving tasks [21]. This autonomy increases when students feel that their opinions are valued and respected, their feelings are taken into consideration, and they are given the opportunity to make their own decisions [22]. Secondly, we find the need for competence through which student feels capable of being able to adequately carry out everything that is proposed, that is, having the sensation of being able to solve those tasks with a high probability of success and efficacy [23]. And, finally, the third BPN is the relationship with others, which is based on the fact that students need to feel connected with their teachers, classmates, and the school itself [24,25]. When students satisfy these BPNs, their engagement, performance, and learning increase [24]. Deci and Ryan [25] understand BPNs as the innate psychological nutrients that are essential for long-term psychological growth, integrity, and well-being.

BPN theory is one of the elements of self-determination theory (SDT) [24,26,27,28]. This theory understands motivation as a continuum in which different levels of self-determination are established. That is, from a higher to a lower degree of self-determination, an athlete’s behavior can be intrinsically motivated, extrinsically motivated, or amotivated. The highest degree of self-determination is possessed by athletes when they are intrinsically motivated, a condition which entails a commitment to sports practice due to the pleasure and enjoyment they obtain from this practice, which becomes an end in and of itself [22,23,24,25,26,27,28,29,30,31,32,33,34,35,36].

Therefore, once the importance of health, the low rates of PA, and the influence that a PE teacher has on the development of motivation and of being able to promote healthy habits have been contextualized, the objective of the present study is to verify the levels of perceived autonomy in Physical Education classes, motivation towards the subject, satisfaction of basic psychological needs, and intention to be physically active among Primary and Secondary Education students. All of this is, in our study, related to BMI.

As for the expected results, the following are hypothesized: (H1) Lower BMI indices will be linked to more satisfactory values in terms of perception of autonomy, motivation, satisfaction of the needs for autonomy, competence, and social relationships, as well as a greater intention to be physically active. In addition, (H2) poses that students at the Primary Education stage will have more adequate values of BMI than Secondary Education students. Furthermore, in terms of gender, it is hypothesized (H3) that boys will also have better BMI values and, hence, following the first hypothesis, more adequate values of psychological well-being with respect to the described variables.

## 2. Materials and Methods

### 2.1. Sample and Design

A descriptive and cross-sectional study was carried out, with a sample selected for convenience and accessibility and made up of a total of 675 students. A series of exclusion criteria were applied to these students—not meeting the response patterns, attending Primary Education or Secondary Education, answering all the questions under study, being within the appropriate ranges using the distance of Mahalanobis as a statistical procedure (taking into account the Body Mass Index as well as the other variables under study)—finally leaving 574 subjects. The participants were all from different areas of the national territory (Region of Murcia, Castilla-La Mancha, Extremadura, and the Valencian Community). The mean age was 13.66 years (SD = 1.96), with a total of 272 male participants (47.4%) and 302 female participants (52.6%). In total, 200 of the participants were students attending Primary Education (34.8%), and 374 of them were attending Secondary Education or Basic Vocational Training (65.22%).

### 2.2. Procedure

First, after carrying out the corresponding bibliographic search, a questionnaire was prepared via Google Docs (URL accesed 25 May 2023 https://forms.gle/3g1igDWkctsNcnCV7) and contact was made with the different management teams of possible study centers and the respective Physical Education teachers. To contact the teachers, advertising was carried out using social networks and contacts from the different researchers. This was accomplished via e-mail, and, later, the main researcher was present when explaining the development of the questionnaire in person or in a blended capacity for the centers outside the territory of the Regions of Murcia and Extremadura via zoom. In turn, an informed consent form was passed around so that it could be delivered to the participants’ parents or guardians, as it was previously indicated that the study would be carried out only (due to the characteristics of the questionnaires) among students from the 4th grade of Primary Education to the 4th grade of Secondary Education. The participants could refuse to be part of the study if they wanted, as indicated in the survey, without any penalty. All the data were treated anonymously by not using identifying codes for the participants, instead having the students themselves completing the questionnaires in their Physical Education classrooms, with a duration between 15 and 25 min depending on their age and level of cognitive development.

Once the procedure was completed, the data were collected and downloaded from the questionnaire prepared using Google Docs for a further analysis. The entire process of development of this work had the approval of the Ethics Committee of the University of Murcia (4447/2023) to ensure that it complied with the guidelines of the Helsinki Declaration on research ethics.

### 2.3. Instruments

A multiple-choice questionnaire composed of five scales was used to detect the following:-Autonomy Support. The Autonomy Support Scale [29] was used. It was made up of 11 items that the participants had to answer about the style of their teacher or coach in class (i.e., “With your explanations, you help us understand what the activities are for what we do”). The sentence used was the following: “In my physical education classes, my teacher…”. The options available consisted of a Likert-type scale from 1 (Surely not) to 5 (Surely yes). The scale obtained an α value = 0.898.-Motivation. The Physical Education Motivation Questionnaire (CMEF) [30] was used, in particular the Spanish version adapted to the context of Primary Education by Leo et al. [31], in order to assess the motivation of the students towards Physical Education classes. The questionnaire consists of a total of twenty items divided into five scales: intrinsic motivation (i.e., “why Physical Education is fun”), identified motivation (i.e., “because I can learn skills that I could use in other areas of my life”), introjected regulation (e.g., “because it is what I have to do to feel good”), external regulation (i.e., “because it is well regarded by the teacher and classmates”), and demotivation or amotivation (i.e., “I don’t know clearly because I don’t like anything “). All of these were answered with a Likert scale from 1 (totally disagree) to 5 (totally agree) and under the premise of “I participate in Physical Education classes...”. The Cronbach’s alpha values obtained were α = 0.879 (intrinsic motivation), α = 0.870 (identified motivation), α = 0.734 (introjected regulation), α = 0.807 (external regulation), and α = 0.752 (amotivation).-Basic Psychological Needs Satisfaction. The Spanish version adapted to the EF [32,33] of the original Basic Psychological Needs in Exercise Scale (BPNES) [34,35] was used. The scale is made up of twelve items (four items per factor) that assess autonomy (i.e., “I think the way physical education is taught is just the way I like it”), competition (i.e., “I think I get better even at tasks that most of my classmates find difficult”), and relationship (i.e., “Relationships with my classmates are very friendly”). Each item began with the phrase “In my physical education classes...”. The responses were collected on a 5-point Likert-type scale ranging from 1 (strongly disagree) to 5 (strongly agree). The Cronbach’s alpha values obtained were α = 0.848 (autonomy), α = 0.892 (competition), and α = 0.865 (relationship with others).-Intention To Be Physically Active. The Measure of Intentionality to be Physically Active (MIPA) questionnaire, created by Hein et al. [36] and adapted to the Spanish context by Moreno et al. [37] with a sample of Spanish adolescents, was used. In addition, it is an instrument that has also been used with Spanish primary school students [38] in a study where the questionnaire’s psychometric properties were analyzed, showing high internal consistency (α = 0.80) and temporal reliability (ICC = 0.79). This is a questionnaire widely used with schoolchildren that consists of various statements which must be answered using a Likert-type scale that ranges from 1 (not at all) to 4 (a lot). The scale had a reliability value of α = 0.798.-Body Mass Index (BMI) Self-Report: it was calculated from the self-reported height and weight of the participants using the formula of weight/height in meters raised to the square.

### 2.4. Statistical Analysis

First, atypical cases were removed from the database by applying the Mahalanobis distance, initially taking into account the BMI and, secondly, including the rest of the variables under study. Next, to analyze the normality of the variables, a Kolmogorov–Smirnov test was performed, showing a non-normal distribution (*p* < 0.05). Then, the asymmetry and kurtosis statistics were analyzed, showing adequate values for all the variables, giving values below two in asymmetry and seven in kurtosis [39]. Next, a correlation analysis was performed using the Kendal Tau b test to verify the correlation between the variables.

Finally, to verify the differences according to the gender and educational stage, the Mann–Whitney U test was performed, taking into account a level of significance of *p* < 0.05. All the analyses were performed using the SPSS version 25.0 software package (SPSS Inc. Chicago, IL, USA).

## 3. Results

### 3.1. Descriptive and Correlation Analyses

The correlation analysis showed statistically significant correlations between almost all the variables. In this sense, Table 1 shows how intrinsic motivation was positively correlated with all the variables except for amotivation. In turn, it is noteworthy that the BMI showed negative correlations with the most self-determined forms of motivation, basic psychological needs, and the intention to be physically active. It only had a positive correlation with amotivation at *p* < 0.001 and was not linked to autonomy support, neither positively nor negatively. Regarding the variables which were not related to each other, we found introjected regulation and demotivation, demotivation and autonomy support, and, as mentioned, BMI and autonomy support.

### 3.2. Analysis According to Educational Stage

Secondly, the differences according to the educational stage in the variables under study were verified. Thus, in Table 2, it can be seen that there were statistically significant differences at *p* < 0.01 for the majority of the variables, with the Primary Education group having higher values in all the types of motivation, the satisfaction of basic psychological needs, and the perception of support of autonomy. In the same way, it can be seen how the BMI was the only variable that obtained a significantly lower value, adding to this result a greater intention to be physically active for this group (*p* < 0.05; Z-score = −2.067). No significant differences were found in the amotivation variable between the groups.

### 3.3. Analysis by Gender

Finally, the differential analysis based on gender (Table 3) did not show significant differences in any of the variables. It is noteworthy in this regard that no trend was observed using parametric procedures and that only the BMI obtained a value close to significance (*p* = 0.088), followed by autonomy support (*p* = 0.103), both being slightly higher values in girls.

## 4. Discussion

The objective of this research was to find out the relationship between BMI, perceived autonomy in PE classes, motivation towards this subject, satisfaction of BPNs, and intention to be physically active in students attending Primary and Secondary Education, differentiating the results based on educational stage and gender. The main findings that have been reached are discussed below.

First, it is worth noting the negative correlation of the BMI with self-determined motivation, identified, introjected, and external regulations, satisfaction of basic psychological needs, and intention to be physically active. Conversely, the BMI did report a positive correlation with amotivation. This result can be interpreted by focusing on students with higher BMI levels, mainly overweight or obese, who are the ones who show the worst values in motivation, satisfaction of psychological needs, and intention to practice in the future. The association of these types of characteristics with students with higher BMIs was also pointed out in previous studies carried out in educational settings such as Gil et al.’s study [40] on sixth-grade schoolchildren and Royo et al.’s work [41] with school-aged adolescents. Likewise, a greater intention to be physically active among students with lower BMI levels also coincides with previous studies, such as the one carried out by Bardid et al. [42] among schoolchildren. According to Urrutia et al. [43], in their study conducted on high school students, these results could be explained if perceived competence is conceived as a mediator in such a way that a lower BMI would be associated with a greater perceived competence and the latter, in turn, with a greater intention to practice the activity [44]. Stodden et al. [44] argued that the development of motor skill competence is a primary underlying mechanism promoting engagement in physical activity. It should not be forgotten that the results of the research conducted by Weiss and Amorose [45] already indicated that perception of motor competence is an important variable for explaining the processes of motivation towards physical-sports practice, which, in turn, influences BMI levels.

The association between a higher BMI, dissatisfaction of the BPNs, and the worst values in self-determined motivation in students had also been previously pointed out by Sáenz-Álvarez [46] as a field of interest that has not been studied until now and requires greater research due to its enormous importance in the educational field.

Secondly, as far as notable findings are concerned, the results have reported significant differences with regard to the educational stage. Lower values in the BMI, higher self-determined motivation, intention to be physically active in the future, feeling of satisfaction of BPNs, and perception of support of autonomy on the part of the PE teacher are found in the group of students attending Primary Education compared to those attending Secondary Education. The transition from Primary to Secondary Education has been described as a phase of psychological, biological, and emotional transformation typical of the entry into puberty–adolescence [47], a turning point when the change of educational stage coincides with the incorporation of student into a new educational center, with great environmental and situational changes [48].

The increase in BMI with the passage from childhood to adolescence coincides with the conclusions of Navarro-Solera et al. [49] who postulate, in samples taken from the same environment as this research, that these differences could be due to a lesser adherence to the Mediterranean diet as the age of students increases in the different educational stages, due to the change in eating habits and customs. In other words, good eating habits are being lost as students grow older and adopt more autonomy and independence when it comes to eating, which highlights the need to re-educate eating habits in children and adolescents.

The reduction in the degree of satisfaction of the BPNs and the level of self-determination in the motivation of older students is in line with the postulates of the theory of self-determination [24] and, empirically, with the results obtained recently by Manzano [50] in primary and secondary school students. The inclusion of the intention to be physically active in the future reinforces the conclusions of a previous study of student profiles carried out by Granero-Gallegos et al. [51], which shows that students with higher motivation are those who practice more physical activity and those who are more likely to acquire and adhere to leisure-time sports practice habits.

Finally, another noteworthy finding would be that there were no significant differences in the variables analyzed in terms of gender. The literature is not clear in this regard, finding both results in adolescent students which confirm those found in this study [52,53] and others, also carried out among similar samples, which refute them [11,54]. With respect to the BMI, according to the WHO [11], there are differences between the genders in terms of healthy lifestyle habits that have an impact on the development of a higher BMI among boys. The differences more directly linked to the educational future of young people, such as, in the case of this research, motivation, BPN, and support of autonomy by teachers, may be due, as Cerezo and Casanova [54] and Soos et al. [55] point out in studies conducted on adolescents, to the perception that boys and girls have of their teachers in PE classes and the differential treatment they receive from them. This would be a research line to develop in the future.

However, in the interpretation of these results, a series of limitations should be considered, such as, for example, the way of measuring BMI based on self-reported data instead of a measurement, generating data which may differ slightly from reality [56]. In this same sense, BMI as an indicator of obesity or overweight can also be questionable if body composition is not analyzed. In addition, although the administration of the questionnaires was carried out in the presence of a researcher who was able to resolve questions, it must be admitted that the length of the questionnaires could generate a tiring effect, especially among younger students, which could affect the marked answers. Moreover, the questionnaires used do not allow one to delve into the reasons why certain answers were marked. Finally, it is important to say that physical activity intention is a good marker to assess the intention to practice sports, but it would have been more objective to measure physical activity levels.

In addition, there exist some methodological limitations that could be also considered as prospects for future research, and they are as follows: (a) since this was a non-experimental study, it did not modify any variable in the real setting where it was carried out so as to improve the values of some variables; (b) akin to any research that administer questionnaires as a means for gathering data, the resulting information could have been mediated by social desirability (even if anonymity was ensured); and (c) the sample selection was not based on a non-randomized process. In potential future research, assessing the model with different educational samples as well as using objective methods for measuring BMI should be considered.

## 5. Conclusions

Based on what has been indicated in the previous sections, it is concluded that there is a positive correlation between intrinsic motivation and the rest of the variables except for amotivation, in which case the correlation, although still statistically significant, is negative among this sample. Furthermore, the BMI is negatively correlated with the most self-determined forms of motivation, basic psychological needs, and the intention to be physically active among this sample. This supposes the partial acceptance of H1 to the extent that the expected correlations are met, but they are not shown between the BMI and the satisfaction of basic psychological needs. On the other hand, the presence of significantly higher values in all the variables studied related to psychological well-being in the group of Primary Education students implies compliance with H2. Finally, the absence of differences based on gender implies rejecting H3.

In this way, the path is opened for future research that includes qualitative information collection techniques in such a way that mixed-type designs are carried out to allow the triangulation of information, providing greater robustness to the research developed. In this same line, the measurement of height and weight, or the inclusion of body composition, using direct techniques is suggested. The possible connection of the variables studied with academic performance is also a field of knowledge attracting growing interest that still requires a larger body of research, as well as possible differences between different cultures and/or samples from different countries. This research also proposes to carry out studies that allow classifying subjects into different profiles based on the variables of BMI, motivation, physical activity intention, satisfaction of basic psychological needs, and autonomy support on the part of Physical Education teachers. And, finally, teacher training plans could be applied for the acquisition of strategies on how to support student autonomy and improve their satisfaction in Physical Education classes, implementing intervention programs with adolescents and, thus, evaluating whether these improve the students’ levels of physical activity in their free time and their BMI.

## Figures and Tables

**Table 1 behavsci-13-01015-t001:** Descriptive and correlation analyses of the variables under study.

	M	SD	A	S	1	2	3	4	5	6	7	8	9	10	11
Intrinsic Motivation (1)	3.943	1.011	−1.000	0.422	1	0.595 **	0.404 **	0.286 **	−0.151 **	0.396 **	0.502 **	0.548 **	0.496 **	0.394 **	−0.093 **
Identified Motivation (2)	3.6838	1.0556	−0.607	−0.455		1	0.402 **	0.278 **	−0.143 **	0.346 **	0.510 **	0.455 **	0.418 **	0.429 **	−0.094 **
Introjected Regulation (3)	3.0814	1.0662	0.006	−0.741			1	0.439 **	0.04	0.232 **	0.368 **	0.337 **	0.263 **	0.259 **	−0.073 *
External Regulation (4)	2.8833	1.13454	0.163	−0.915				1	0.173 **	0.117 **	0.277 **	0.235 **	0.175 **	0.209 **	−0.071 *
Amotivation (5)	1.6686	0.90964	1.54	1.991					1	−0.116 **	−0.058	−0.112 **	−0.141 **	−0.048	0.099 **
MIPA (6)	4.084	0.85679	−0.831	−0.043						1	0.333 **	0.466 **	0.335 **	0.268 **	−0.068 *
Autonomy (7)	3.1681	1.03349	−0.165	−0.669							1	0.513 **	0.448 **	0.506 **	−0.072 *
Competence (8)	3.7744	0.94079	−0.659	−0.186								1	0.518 **	0.338 **	−0.128 **
Relationship (9)	3.8885	1.01386	−0.894	0.113									1	0.337 **	−0.120 **
Autonomy Support (10)	3.6771	0.96136	−0.659	−0.165										1	−0.044
BMI (11)	19.579	3.32153	0.748	0.673											1

Note: BMI = Body Mass Index; MIPA = Measure of Physical Activity Intention; M = mean; SD = standard deviation; A = Asymmetry; S = Weakness; * = *p* < 0.05; ** = *p* < 0.001.

**Table 2 behavsci-13-01015-t002:** Differential analysis according to the educational stage.

	Primary Education		Secondary Education or Basic Vocational Training			
	M	SD	M	SD	Z-Score	*p*
Intrinsic Motivation	4.35	0.82	3.72	1.04	−7.6888	0.000 **
Identified Motivation	4.11	0.87	3.46	1.08	−7.221	0.000 **
Introjected Regulation	3.42	1.03	2.90	1.04	−5.426	0.000 **
External Regulation	3.19	1.19	2.72	1.07	−4.599	0.000 **
Amotivation	1.69	0.98	1.66	0.87	−0.948	0.343
MIPA	4.21	0.78	4.02	0.89	−2.067	0.039 *
Autonomy	3.49	0.91	2.99	1.05	−5.405	0.000 **
Competence	4.05	0.85	3.63	0.95	−5.278	0.000 **
Relationship	4.24	0.81	3.70	1.06	−6.251	0.000 **
Autonomy Support	3.90	0.77	3.56	1.03	−3.469	0.000 **
BMI	18.25	2.76	20.29	3.38	−6.851	0.000 **

Note: R = regulation; MI = intrinsic motivation; BMI = Body Mass Index; MIPA = Physical Activity Intention Measure; M = mean; SD = standard deviation. * = *p* < 0.05; ** = *p* < 0.001.

**Table 3 behavsci-13-01015-t003:** Differential analysis according to gender.

	Boys		Girls			
	M	SD	TO	S	Z-Score	*p*
Intrinsic Motivation	3.97	1.01	4.01	1.04	−0.682	0.495
Identified Motivation	3.69	1.05	3.76	1.06	−0.927	0.354
Introjected Regulation	3.13	1.06	3.10	1.12	−0.309	0.757
External Regulation	2.90	1.09	2.85	1.22	−0.530	0.596
Amotivation	1.60	0.84	1.64	0.91	−0.086	0.931
MIPA	4.17	0.83	4.07	0.84	−1.229	0.219
Autonomy	3.11	1.02	3.23	1.07	−1.386	0.166
Competence	3.87	0.90	3.76	0.96	−1.061	0.289
Relationship	3.97	0.95	3.97	1.01	−0.304	0.761
Autonomy Support	3.62	0.96	3.77	0.92	−1.630	0.103
BMI	19.20	3.40	19.64	3.22	−1.704	0.088

Note: R = regulation; MI = intrinsic motivation; BMI = Body Mass Index; MIPA = Physical Activity Intention Measure; M = mean; SD = standard.

## Data Availability

The data presented in this study are available on request from the author David Manzano-Sánchez. The data are not publicly available due to privacy issues.

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
