# Peer review of "Body Mass Index: Influence on Interpersonal Style, Basic Psychological Needs, Motivation, and Physical Activity Intention in Physical Education—Differences between Gender and Educational Stage"

_behavsci, 2023, doi:10.3390/bs13121015_

Round 1

Reviewer 1 Report (Previous Reviewer 2)

Comments and Suggestions for Authors

The authors have reflected on and revised almost all of the reviewer's points. The only thing that has not changed is the suggestion of calculating reliability based on McDonald's omega, but it is something of no substance. The current version of the manuscript is perhaps publishable.

Author Response

Thank you very much for your review. We have considerate the use of McDonald’s omega, but the results were similar with both tests.

Reviewer 2 Report (New Reviewer)

Comments and Suggestions for Authors

Dear Authors, the manuscript is well done, with a good structure, content and research methodology. Also, the discussions are coherent and conclusions are in relation with results.

As recommendation, I suggest to replace "sex" with "gender".

Author Response

Thank you very much for your review. The term has been reviewed and modified.

Reviewer 3 Report (New Reviewer)

Comments and Suggestions for Authors

General comments:

The study is original and relevant, it refers to a topic that is still little studied and has potential to contribute to research and application. However, the manuscript needs some adjustments and clarifications as follows.

Introduction:

More organized writing of the introduction...

Overview and definition of the problem. What's the problem?

Approach to the exposures and their relationships with the problem;

Approach to the outcome and its relationship with the problem;

Justification and importance for carrying out the study;

Expected results.

Methods:

Adequate description of the procedures.

Did the researchers not assess the PA level of the sample?

Since there is a relationship between PA and the variables analyzed, that is, more active subjects have better BMI and BPN values, the results may be influenced by the level of physical activity of the students. This needs to be discussed.

Results:

The results section needs a descriptive table that shows the data according to the different categories of analysis, that is, by educational level and gender.

Discussion:

The authors need to further expand the discussion on the findings and not just indicate whether it is in according with other studies.

Author Response

Thank you very much for your considerations. We have worked to be able to attend to all aspects.

So as to provide a response for each one, we keep the used structure:

Introduction:

More organized writing of the introduction...

The introduction has been modified so that the organization and concretion is better.

Overview and definition of the problem. What's the problem? Approach to the exposures and their relationships with the problem; Approach to the outcome and its relationship with the problem.

The problem has been defined and the approach to the problem has been stated.

Justification and importance for carrying out the study; Expected results.

The importance of the study and the relationship of the results with the study problem has been justified.

Methods:

Adequate description of the procedures.

Did the researchers not assess the PA level of the sample? Since there is a relationship between PA and the variables analyzed, that is, more active subjects have better BMI and BPN values, the results may be influenced by the level of physical activity of the students. This needs to be discussed.

The importance of measuring physical activity to draw new conclusions has been indicated in the limitations section and has been discussed in this section.

Results:

The results section needs a descriptive table that shows the data according to the different categories of analysis, that is, by educational level and gender.

The results have been reviewed. But in tables 2 and 3, you can see the data broken down by gender and by stage depending on the variables under study. If you require any additional information, we are at your disposal.

Discussion:

The authors need to further expand the discussion on the findings and not just indicate whether it is in according with other studies.

The discussion has been deepened to discuss the results obtained.

Thank you for your comments.

Round 2

Reviewer 3 Report (New Reviewer)

Comments and Suggestions for Authors

I would like to congratulate the authors for the topic covered due to its importance for the development of children and adolescents. Furthermore, it adds information that can be used in the daily practice by physical education teachers.

The authors justified and answered all comments. the study is now suitable for publication.

Yours sincerely.

This manuscript is a resubmission of an earlier submission. The following is a list of the peer review reports and author responses from that submission.

Round 1

Reviewer 1 Report

Comments and Suggestions for Authors

The manuscript's structure is fine, the work is quite sufficiently described, the language use is good. There are some areas that I would like to see even further improved:

* * *

Usually there is no need to use markers like (1) Background; (2) Methods; (3) Results in the abstract section of the paper. I personally would remove them, but I am not insisting that they need to be removed. Otherwise, the abstract is nice. 

*

Introduction is written finely, but it needs to discuss BMI in more details. In particular, BMI should be defined, its normal values and its dynamics throughout person's life should be explained. Also, surely there is some literature that has examined relationships between BMI, PE and relevant aspects. This literature should be discussed also, and more important papers on this matter should be shortly described.

*

Sample - please describe how did you approach/find your participant PE teachers (through social media, through schools, via public ads, or some other methods). Also, describe if the participant students had the right to refuse participating in the study, and if there were any consequences for them if they did so. Discuss - Did you collect data that allows identify any of the participants? Did you collect any sensitive information? If so then also describe how were students' interests protected (after all, they belonged into vulnerable age group). Describe in expressis verbis, who filled in the questionnaires - the students, their PE teachers or their parents. Also, where was the questionnaire filled in (at home, at school), how much time did it take.

If you based your results on self-reported data, please also discuss the possible self-report bias (e.g. https://journals.sagepub.com/doi/full/10.1177/2042533313514048 - "Comparison of bias resulting from two methods of self-reporting height and weight: a validation study"). This can be discussed also in the Discussion section. 

*

Discussion and Conclusion 

Please discuss the limitations of your study here. Please discuss future directions in more details.

Reviewer 2 Report

Comments and Suggestions for Authors

Dear authors, some errors seen, or suggestions for improvement are shared:

Abstract

19. The meaning of the M is not indicated. It is supposed to be the age, but is better if it is specified.

25. It should be avoided the use of “a more positive psychological profile”, due to is imprecise and axiological.

26-28. That is not a scientific conclusion. It should be concluded something about the scientific goal. Probably the author could already affirm this before doing this study, and therefore it is not a conclusion (but almost a starting premise, or a social objective of the researcher).

Introduction

40. Neither of the two studies cited affirm such a thing, and neither can they affirm it. The author here makes a causal claim, and reinforces it with correlational studies.

44-46. These types of sociopolitical (axiological) statements should be avoided. Science describes and explains how the world is. Science does not tell us what to do in life. Therefore, if this is intended to be a scientific study, guidance should be avoided. Readers will already know what to do (perhaps) with the findings of this study.

68. It's the other way around. ADT is the macrotheory, within which there are several (such as BPN)

75-95. These two paragraphs are full of substantiated truths, but they do not justify the objective of this correlational study. The relevance of the teacher is not necessary to develop so much here, since it is not one of the variables considered in this study. It is necessary to delve deeper into the variables that will be measured, and the relationship between them.

99. There seems to be a double space. It must be well justified that BMI is a marker of physical or psychological health.

104. BMI has been criticized as a measure that allows drawing conclusions regarding well-being. Justify yourself better, please.

113-114. Why has this sampling criterion been chosen? What is the population (not the sample)? How do you know what selection biases there may be? If a good sampling technique is not used, nothing can be said except about the sample itself.

135. Congratulations on having passed through the regional Ethics Committee

 156-159. Better results have been found measuring reliability with McDonald's omega (instead of alfa's Cronbach).

McDonald's omega. Please see Hayes & Coutts (2020) and McDonald's (1999).

References:

Hayes, A.F., & Coutts, J.J. (2020). Use omega rather than Cronbach's alpha for estimating reliability. But…. Communication Methods and Measures, 14(1), 1–24.

McDonald, R.P. (1999). Test theory: A Unified Treatment. Mahwah, NJ: Lawrence Erlbaum. https://doi.org/10.4324/9781410601087

178-179. The name of this variable should then be changed to “BMI perception”. In this case, the way it is measured affects the construct itself.

189. Sex and gender are not the same. If the author/authors want to refer to the biology of the participants, they should put "sex". Gender is quite ambiguous and unspecific, as current sociology shows.

216. Idem. Furthermore, it must be justified why a difference is made by sex in this type of study. The review knows that it is a tradition to separate by sex, but it is not always relevant at a scientific level, and therefore a difference is introduced that is pre-scientific or extra-scientific. It should be better justified.

271-277. This explanation of the previous literature is not really a discussion (it does not explain the results of the study),but should go in the introduction as a justification.

291-300. In all statements of the conclusion "in this sample" must be added. Otherwise, you are committing a generalization bias. The method limits what can or cannot be concluded.

The scientific effort of the author or authors is appreciated. However, there would be many improvements to be made.

Reviewer 3 Report

Comments and Suggestions for Authors

In my opinion, the article lacks novelty. In my view, although we have various specialized tools to measure in-body components, relying solely on the BMI variable appears outdated in terms of research methodology. BMI alone is no longer sufficient to accurately describe one's health. Numerous studies have demonstrated that a high BMI may not necessarily indicate obesity but could instead be attributed to increased skeletal muscle mass. Additionally, psychology research has consistently shown that self-determination plays a pivotal role in achieving success across various domains, including maintaining good health, following a proper diet, and engaging in regular physical activity.